# Self-Assessment Method for Sustainability Implementation in Product Innovation

**Jesko Schulte ***  **and Sophie Isaksson Hallstedt**

Department of Strategic Sustainable Development, Blekinge Institute of Technology,
SE-37179 Karlskrona, Sweden; sophie.hallstedt@bth.se
**\*** Correspondence: jesko.schulte@bth.se; Tel.: +46-455-385-519

**Abstract:** Companies, striving towards an effective and systematic integration of a strategic sustainability perspective in product innovation, need to treat the implementation of necessary processes and tools, as well as their continuous improvement, as a project in itself. An efficient way to measure the current sustainability implementation level in the organization, as well as guidance for progression, is required. To meet this need, a novel self-assessment was developed, which provides companies with a tool to assess and visualize their current capabilities in relation to key elements for successful sustainability integration in the product innovation process. It includes a scale of different sustainability implementation levels to support building a roadmap for systematic implementation, and to measure progress over time. This research is based on results from previous descriptive work within the area of sustainable product development and learning from applying strategic and tactical assessment tools for eco-design and sustainability maturity. Besides the contribution to practice, this study also contributes to knowledge by specifying detailed aspects for each key element that must be considered to guide sustainability integration. Also, insights from applying different existing tools in real cases are provided. The newly-developed self-assessment method was applied and validated at two case companies. Independent and continuous use of it by the companies beyond this particular study indicate the practical value of the method.

**Keywords:** application; case study; corporate sustainability; implementation; maturity; measure; sustainable product development

## 1. Introduction

Both natural and the social systems are currently on downward spirals: pollution and encroachment are undermining nature's capacity to provide a stable as well as beautiful environment for life on Earth [1] and social capital is eroding in many places, leading to increasing transaction costs [2]. Product development and manufacturing companies play a key role in facilitating the transition towards a sustainable society [3].

Several mechanisms work as driving forces for companies to change towards a more sustainability-oriented state [4]. First and foremost, companies start to acknowledge the self-benefit of enhanced capabilities for sustainability integration through increased competitiveness [1]. Here, capability refers to which and how method and support systems are used, and which and how sustainability-related decisions are taken during the product innovation process. According to Roozenburg and Eekels [5], the product innovation process includes both product development, which in turn consists out of product planning and strict development, as well as realization. The self-benefit can manifest itself through, for example, new innovations that provide more customer value, the attraction and maintenance of top talent employees, a strong brand and business image, being ahead of legislation, more motivated and loyal employees, lower operating and credit costs,

and lower vulnerability to sudden market changes [6,7]. In short, engaging in sustainability is a way to exploit business opportunities and avoid long-term negative consequences, which means that it is smart risk management [8].

However, even companies that are aware of the necessity and benefits of increasing their capabilities for sustainability integration are struggling with translating their ambitions into systematic and strategic change in their organization and practices [9]. One of the main challenges is the uncertainty regarding which practices and aspects that are relevant for sustainability integration in the product innovation process, as well as how these aspects can be systematically improved. Clear objectives, success criteria, proper planning and resourcing, as well as monitoring and control are required. For this to be achieved, a thorough understanding of the company's current capabilities and a clear definition of the intended outcome are necessary. This means that companies need to be able to assess present maturity and capability, and to define progress towards higher levels of sustainability implementation maturity. Based on this, the following research question is addressed:

- How can the maturity of company practices be assessed and visualized to efficiently and systematically guide sustainability implementation in the product innovation process?

To answer this question, an interactive qualitative research approach [10] was applied. It follows the structure of the Design Research Methodology [11], including a descriptive study at case companies, a prescriptive study, and a descriptive validation study. By answering the research question, the main contribution of this paper is twofold: firstly, through the practical application of several existing methods for maturity assessment, this study contributes to knowledge by providing experiences on their usefulness and limitations. Furthermore, detailed aspects that companies need to consider in their sustainability implementation journey were identified. Secondly, based on these insights and a backcasting perspective [12], a new method for assessing an organization's current capabilities, in relation to key elements for successful implementation of sustainability in the product innovation process (further explained in Section 2.1), was developed and tested in two product development and manufacturing companies.

*Outline*

In the next section, the literature on decision support and maturity models for sustainability integration is reviewed, focusing on the concepts and tools that were utilized in this study. This is followed by Section 3, which describes the research approach that was used to provide an answer to the research question. In Section 4, the results are presented and discussed, starting with findings from exploratory and descriptive studies and the application of existing tools, before an improved self-assessment method is proposed and validated. Conclusions, as well as limitations and future research directions, are stated in the Section 5.

## 2. Review of Related Work

### 2.1. Decision Support for Sustainability Implementation

Sustainability implementation refers to the practical usage and application of tools, methods, processes, approaches, practices, etc., that aim to improve an organization's contribution to sustainable development. An important means for implementation is to use decision support [13] and there is a wide range of sustainability-oriented methods and tools for decision support in product development [14–16]. Most methods and tools designed to support sustainability considerations in the product innovation process, i.e., integration, have however, a low level of practical applicability, i.e., implementation [17].

As Baumann et al. [18] pointed out in 2002, there is too much development of new tools within the area of sustainable product development and still too little utilization and improvement of existing research and tools. Following their advice, this study actively built on, tested and evaluated several

existing tools for sustainability maturity assessment, to see if and how they could be further advanced. This resulted in the development of an improved self-assessment approach that leverages on the strengths of existing tools. Also, this study utilizes and expands on existing knowledge and concepts in the field, as well as previous exploratory and descriptive research especially regarding key elements for successful implementation of a strategic sustainability perspective in the product innovation process [19]. These were used for structuring information and data collection, as well as for guiding the prescriptive research. The key elements are based on the conceptual understanding of sustainability of the Framework for Strategic Sustainable Development [20]. In short, the eight key elements are [19]:

1.　Ensure organizational support from senior management;
2.　Efficiently introduce a sustainability perspective early in the product innovation process;
3.　Utilize knowledge and experience of procurement staff in the earliest phases of the process;
4.　Include social aspects across the product life cycle and its value chain;
5.　Assign responsibility for sustainability implementation in the product innovation process;
6.　Have a systematic way to share knowledge and build competence in the sustainability field to inform decisions taken in future product development projects;
7.　Utilize tools for guiding decisions as a complement for assessment tools; and
8.　Utilize tools that incorporate a backcasting perspective from a definition of success.

The key elements are, however, too general to be operationalizable. Therefore, this study aims to add detail and to make the key elements applicable in practice, thereby providing companies with a tool to assess and visualize current capability levels and to systematically guide sustainability implementation in the product innovation process.

### 2.2. Sustainability Maturity Models

Several sustainability maturity models have previously been developed in research and practice, ranging from very simple to highly sophisticated ones. Some of these models take an overarching perspective and are applicable for most types of corporations, e.g., [21–24], while others focus on specific industries, e.g., [25], or business aspects, such as new product development [26], eco-design [27], or supply-chain sustainability [28]. Several business excellence models, indexes, and standards also incorporate some sustainability aspects and can be used for self-assessment purposes [29]. The existing maturity models that were applied in this study are described in more detail below.

Willard [30] described five levels of sustainability integration, based on case studies of front-running companies. The stages range from pre-compliance to integrated strategy and purpose, see Figure 1. According to Willard [7], if the transition from stage 2 to stage 3 can be described as a step, the transition from stage 3 to stage 4 is a leap. Therefore, stage 3 can be further divided into four sub-stages. In this case, the stages do not have to be achieved in any specific order, but they are all touched upon sooner or later on the way from stage 3 to 4.

- Stage 3.i: Improve company eco-efficiencies and sustainability brand. Same products and/or services in the same processes.
- Stage 3.ii: Improve supply chain conditions and footprints. Business to business, support suppliers to achieve stage 3.
- Stage 3.iii: Create new eco-effective products, services and leases. Redesign, green innovations, leases instead of selling, take product back after end-of-life.
- Stage 3.iv: Embed sustainable governance. Sustainability into decision making, policies, culture. Transparently reporting about contribution to a sustainable global economy, society and environment.

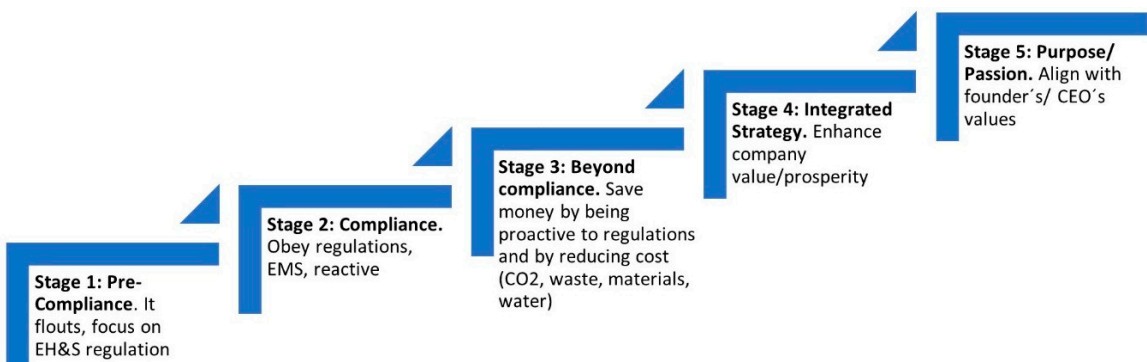

**Figure 1.** The five levels of sustainability compliance [7].

Baumgartner and Ebner [31] explored profiles of company sustainability strategies with the aim of helping companies that are already working with sustainability to assess whether they are consistent in the implementation of a certain sustainability strategy. As part of this work, they identified relevant sustainability issues that such a strategy should incorporate. The identified aspects are grouped according to the three dimensions of corporate sustainability: ecological, social (internal and external) and economic dimensions. In total, Baumgartner and Ebner [31] included 19 aspects, each of which could occur at one of four different maturity levels: beginner, elementary, satisfying, or, sophisticated/outstanding level. A company's pattern of maturity levels in regard to the aspects would then be compared to five distinct sustainability strategies: transformative extroverted strategy, conservative strategy, visionary strategy, conventional visionary strategy, or, systemic visionary strategy. In order to assess the corporate sustainability strategies, templates based on the identified sustainability aspects at different maturity levels were developed and presented by Baumgartner and Ebner [31].

The Eco-design Maturity Model (EcoM2) was developed by Pigosso et al. [32] to help companies implement and manage eco-design in a systematic and consistent way. The method aims at improving the processes related to product development within the company, as good processes lead to good products. As a first step, the current maturity profile of the company's eco-design implementation is assessed. Based on that, the most suitable eco-design practices and ways of improvement are identified and roadmaps for the next steps for implementation can be made. The model works in a loop similar to the PDCA (Plan-Do-Check-Act) to ensure continuous improvement. The eco-design maturity level is defined as a combination of evolution levels and capability levels. There are five evolution levels, which are similar to those of Willard. Each evolution level is connected to management practices, which have one of five capability levels, ranging from incomplete, ad hoc, and formalized levels, to controlled, and, finally, improved levels. The evolution and capability levels are then combined, which results in the maturity level. The EcoM2 is currently further developed to also include practices for social innovation and Product Service Systems [33].

The Template for Assessing Decision Systems (TADS) is a method that aims to give an overview of the strategic capability of company decision systems, both in general and for sustainability in specific [13]. The purpose is to identify a list of key improvements for how a company could integrate sustainability in its strategic decision system. This is done by using template questions that address the five levels of the Framework for Strategic Sustainable Development [20], in relation to the company's decision systems, see Table 1.

None of the mentioned maturity models is today widely used in practice and experience on their applicability and usefulness is limited.

**Table 1.** Template questions for assessing the strategic capability of company decision systems in general and for sustainability. Based on [13].

| Levels of Generic Assessment Framework | Template 1 | Template 2 |
|---|---|---|
| | **Assessing Company Decision System—for General Strategic Capability** | **Assessing Company Decision System—for Strategic Sustainable Development Capability** |
| 1. System | How does the company describe its business idea and operations in relation to key stakeholders? | How does the company describe its business idea and operations in relation to ecological and social sustainability and to stakeholders globally? |
| 2. Success | How, if at all, does the company define its long-term success? | How, if at all, is global sustainability integrated in the company's long-term success definition? |
| 3. Strategic Guidelines | How, if at all, does the company use overarching strategic guidelines for planning towards success in general? | How, if at all, does the company integrate sustainability in overarching strategic guidelines? |
| 4. Actions | How, if at all, are decisions in practice made in line with strategic guidelines towards the company's long-term definition of success? | How, if at all, are decisions in practice made in line with strategic guidelines towards the company's long-term definition of success? |
| 5. Tools | How, if at all, are decisions justified and monitored by suitable methods, tools and concepts? | How, if at all, are decisions justified and monitored by suitable methods, tools and concepts? |

## 3. Methods

The research is based on an interactive qualitative research approach [10], which structure is guided by the Design Research Methodology [11]. It shows exploratory and descriptive findings from multiple empirical case studies, followed by a prescriptive study and a descriptive validation case study, see Figure 2. In research stage 1, the aim was to assess the capabilities for sustainability implementation at the case companies. Guided by the research question, a questionnaire and interviews were conducted and four existing methods for measuring sustainability maturity were applied. It is at this point that a gap was identified between company needs and how the existing maturity models work and what results they provide. The process of assessing the companies' current state was also time and resource demanding and not feasible to be applied by companies in general. This presented the motivation for research stage 2, in which the findings and experiences from research stage 1 were utilized. As a result, a new self-assessment method was developed that provides a quicker and easier way that companies can apply without external support. A first validation of the new method was conducted in research stage 3.

In total, four multinational case companies, all doing product development and manufacturing, participated in the study, see Table 2. Similar to Høgevold et al. [9] and Dangelico and Pujari [34], purposeful sampling was applied, and the companies were selected because they have started to work more actively with sustainability aspects beyond the level of mere compliance, without being companies that define themselves solely from a sustainability perspective. This makes them both relevant for understanding how companies work with sustainability in product innovation, and representative for a broad spectrum of businesses, which increases the external validity of the findings. Company A is a medium-sized (about 100 employees) lamp manufacturer with their competitive edge being smart lighting solutions with environmental-friendly long-life lamps. Company B is a large (about 2000 employees) manufacturer of jet engine components in the aerospace industry. Company C is a large (about 4000 employees) machine manufacturer. Company D is a research center with about 250 employees for a multinational company in the manufacturing industry. The studied company sites were located in Sweden. The exploratory and descriptive research stage involved all four companies to gather broad information and to get a more representative picture of the current state of sustainability capabilities in companies, as well as to test existing tools in different contexts. As gathering in-depth insights and company interaction were the primary purpose of the

following research stages, the focus was on three (prescriptive research stage) and then two (validation research stage) case companies.

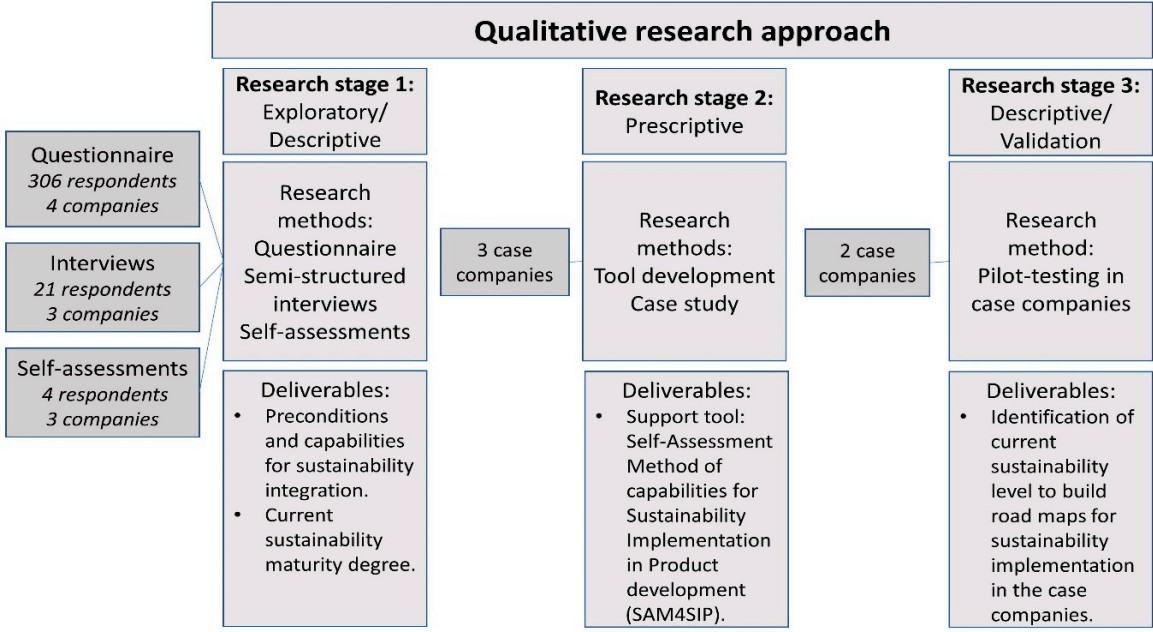

**Figure 2.** Illustration of the research process with research stages, methods and deliverables.

**Table 2.** Case company characteristics and participation in the different research stages.

| Company | Turnover, k€ | Employees | Questionnaire Respondents | Interview Respondents | Self-Assessment | SAM4SIP Validation |
|---|---|---|---|---|---|---|
| A | 75,000 | 100 | 16 | 5 | 1 | - |
| B | 800,000 | 2000 | 69 | 8 | 4 * | 1 group (3–5 persons) |
| C | 2,750,000 | 4000 | 38 | 6 | 1 | 1 group (3 persons) |
| D | 50,000 | 250 | 183 | - | - | - |

* The different parts of the templates on sustainability strategies were at company B filled in by four selected persons who were considered to have the right competence and company experiences for the task.

## 3.1. Research Stage 1: Exploratory and Descriptive Studies

In research stage 1, several methods were triangulated to investigate employees' perceptions of the companies' current work with sustainability aspects in relation to the product innovation process. The studies were conducted partly in parallel.

### 3.1.1. Questionnaire to Identify Common Preconditions and Capabilities for Sustainability Integration

The first empirical study included all four case companies and aimed at gathering survey data from a large group (n = 306) of engineers and product developers, with regard to the ability of managing future needs for sustainability at the company. A web-questionnaire was designed to identify common preconditions and capabilities for sustainability integration in the product innovation process. In total, 22 questions (Appendix A) were formulated to cover the following areas (i) general information about the respondent such as role and working experience, (ii) the importance and prioritization of sustainability integration as well as motivations for doing this, (iii) decision making, (iv) decision support, (v) challenges, and, finally, (vi) improvement suggestions. Different types of open and closed questions were combined in order to gather both quantitative and qualitative data and to achieve synergistic effects and a balance between the weaknesses of each type of questions and scaling techniques [35]. Pilot testing was done prior to the study at one company, which led to improvements of the questions.

The recipients of the questionnaire (523) were selected in a non-probabilistic way, in consultation with the companies. The goal was to include as many suitable people in the study as possible, to get representative results and to avoid selection bias. The target group consisted of the persons with roles in product development and technology development such as product developers, project managers, engineers, purchasers, environmental and sustainability engineers and managers.

Within-case analysis was performed prior to searching for cross-case patterns. In this way, the necessary depth of understanding and familiarity with each case as a stand-alone entity was acquired, before comparing results from different cases [35]. Because of the mainly explorative nature of the study, conventional content analysis [36] was deployed in order to avoid preconceived categories in the coding process. Instead, categories were derived inductively [37], directly from the data.

### 3.1.2. Interviews to Investigate Companies' Current Capabilities in Relation to the Eight Key Elements

The second empirical study featured in-depth semi-structured interviews at companies A, B and C, as a complement to the questionnaire. The purpose was to investigate the companies' current capabilities in relation to the eight key elements (presented in Section 2.1). Furthermore, the purpose was to cover some details regarding the current situation in managing sustainability issues in the work of the companies and to identify some challenges and improvement potentials. The interview questions, which were derived directly from the eight key elements, are presented in Appendix B. Five to eight employees from each company (in total 19 people) were interviewed. The employees had an average of more than 10 years of working experience at these companies. Interviewees were selected to provide input from roles that influence or take decisions regarding sustainability issues early in the product innovation process, such as product developers, project managers, chief engineers, purchasers, environmental and sustainability engineers and managers, and, production engineers and managers. Interview sessions were recorded and transcribed by the researchers. The interview transcriptions were sent back to the interviewees for an opportunity to confirm or correct the texts. In addition to the interviews, steering and product policy documents, support tools, and working processes were studied to get a better understanding of whether and how a socio-ecological sustainability perspective is integrated in the documented product innovation process today.

### 3.1.3. Self-assessment of the Current Sustainability Maturity Degree at the Strategic Level

The third empirical study applied four of the already existing self-assessment methods (described in Section 2.2) to assess the current sustainability maturity degree at the strategic level at the case companies, which is connected to the sustainability implementation capability in the product innovation process: (i) Willard's sustainability compliance ladder; (ii) templates for assessing corporate sustainability strategies by Baumgartner and Ebner [31]; (iii) EcoM2; and (iv) TADS. These methods were selected because they are all generic and applicable in any company, they have guidance or templates available, they have maturity levels defined, they can be used as self-assessment support, and they are based on different approaches with the common aim to support companies to implement sustainability in their organization. The purpose was to test the usability of the tools and to obtain answers on the companies' current sustainability maturity degree. The four different types of methods were used to triangulate and to investigate how the different methods perform, for example, whether they give the same or different pictures of the current sustainability maturity degree, and what the methods' strengths and weaknesses are.

For the method by Baumgartner and Ebner [31], the templates were sent to a senior manager at each company with the request to make a self-assessment by marking the descriptions that best matched the company's current status. For TADS and Willard's sustainability compliance ladder, guided questions were presented to the senior manager before a physical meeting with the researcher. The templates were then filled in by the manager during the meeting.

In the case of the EcoM2, which is an extensive and sophisticated tool, nine questions were formulated with the help of the researcher who developed the EcoM2 [32] that can be used as a

self-assessment tool in order to get an impression of an employee's perception of the company's current capability levels in different areas, see Appendix C. These nine questions were created to capture the aspects of the EcoM2 that are most important and relevant for the purpose and scope of this study.

The different self-assessments were conducted at case companies A, B, and C by persons with key roles (see Table 2). A senior manager at each company was asked to make the self-assessment based on the questions from the different methods. A senior researcher was present during the self-assessment but only had the passive role of clarifying the questions if the interviewee was unsure. After compiling the results, the managers, who did the self-assessment, confirmed or corrected the result.

## 3.2. Research Stage 2: Prescriptive Study

The three studies were then followed by the development of a new method called Self-Assessment Method of capabilities for Sustainability Implementation in the Product innovation process (SAM4SIP), which is the main outcome of this research. The aim was to enable companies to (i) assess and visualize their current capabilities in relation to the eight key elements for successful sustainability integration in the product innovation process (presented in Section 2.1); (ii) identify areas of strengths and improvement possibilities for sustainability implementation; and, thereby, to (iii) clarify strategic socio-ecological sustainability goals; (iv) building a roadmap for efficient and systematic implementation of sustainability, and (v) measure progress over time. This new tool was developed based on the empirical studies that were used to define the relevant detailed aspects of each of the eight key elements as well as the detailed maturity levels for each aspect.

## 3.3. Research Stage 3: Validation

The developed SAM4SIP was tested by two researchers for three of the four companies (A, B and C), using the collected information. The method was then applied and validated at two case companies (B and C) as a test to validate the tool's applicability and useability in relation to the expected aims. At company B, five engineers and research and development (R&D) persons filled in all but two of the eight templates, which had a focus on procurement and organizational support. These two templates were sent to the procurement manager and the environmental manager at the company to grade them as accurately as possible. For company C, four engineers and R&D persons filled in the eight templates. The workshop was divided in three sessions. In the first session two groups with 2–3 users of the SAM4SIP at the two companies were asked to discuss: (i) How did you fill in the templates and why did you choose that way (e.g., roles, in groups or individually etc.)?; (ii) What strengths and weaknesses have you identified for each key element?; (iii) Why do you think you have these strengths and weaknesses?; (iv) What are the next steps to make progression? The second session was organized for each company group to discuss how their roadmap could be improved, based on the results of the self-assessment. In the third session, the company groups were asked to fill in a validation questionnaire, see Appendix D, and to discuss what would be required for the tool to be implemented and used. The questions were derived following guidance on application evaluation by Blessing and Chakrabarti [11], focusing on the assessment of the proximal outcomes, i.e., the outcomes that should be directly affected by the tool. The evaluation aimed at providing insights on three key aspects: (i) whether the support can be used; (ii) whether it directly addresses the factors it is supposed to address; and (iii) whether these factors are are affected as expected.

A success evaluation was not yet possible since it would require a longer time period to be able to assess whether the application of the tool has affected the companies' sustainability implementation as intended.

## 3.4. Limitations

The overall research design, as well as the design of the different research stages, have implications for the trustworthiness of the findings. The main limitation of the chosen design with four case

companies is the limited external validity of the findings. However, based on the aim and scope of this study, acquiring detailed insights with high internal validity was more important than gaining a broader overview with a lower level of detail.

Triangulation and respondent validation were used to achieve a high level of credibility. To increase reliability and mitigate bias, especially observer bias and confirmation bias, several researchers were involved in both data collection and analysis. The reliability of the study might also have been affected by participant bias, as the study participants might have provided answers that they thought the researchers wanted to hear. The social desirability effect might have further affected participants' attitudes towards sustainability. Measures to limit this bias were that participants were assured that their responses would be treated confidentially, and questions were formulated and asked in a neutral way.

## 4. Results and Discussion

### *4.1. Research Stage 1: Current State of Company Capabilities for Sustainable Product Development*

Research stage 1 consisted of exploratory and descriptive studies, mainly based on document review, questionnaire, interviews and the application of existing self-assessment tools. The aim of this stage was to assess the companies' current state in relation to their capabilities for sustainability integration in the product innovation process and to gather inputs on employees' attitudes and perceptions. For a detailed presentation and discussion of the questionnaire results, see also [38].

### 4.1.1. Sustainability Importance, Prioritization and Driving Forces

The questionnaire clearly revealed that respondents at all case companies think that increased capabilities for sustainability integration in the product innovation process are very important (7.6–9.4) for the future of the company (on a scale: 1 not important–10 very important). Long-term competitiveness was the most frequently stated driving force, see Figure 3. Respondents perceive it as a business opportunity and as "*[ . . . ] a necessity for survival of our business and a core aspect of our corporate mission*", arguing that it would both lead to cost reduction and to increased customer value. Other important driving forces were legal requirements and environmental concern, i.e., ethical reasons for taking more responsibility for the planet and future generations. These findings add strength to the argument that there has been an evolutionary shift from ecologic to economic concern as the dominating driving force [9,39]. Reputation and brand were perceived as the main driving force at Company C, while it is not among the top 3 at any of the other companies. Only at one company did respondents perceive that strategic decisions act as strong driving forces. However, the opinions regarding the prioritization levels of sustainability at the strategic and operational levels are significantly lower (5.8–7.7 and 5.8–6.5 respectively; scale 1–10). This discrepancy indicates that the importance of sustainability is recognized but that companies are struggling with translating these long-term requirements into correspondingly high prioritization in the present requirements.

The in-depth interviews regarding key element 1, i.e., top management commitment for sustainability, showed that there is high commitment at the chief executive officer (CEO) level and parts of the senior management group at two of the case companies. The middle-management level is, however, perceived to be generally less committed to sustainability. There are also ambitions to include a sustainability perspective in the product innovation process. The companies are, however, struggling with how to do this and how to achieve a full sustainability perspective that includes the social dimension and that goes beyond energy efficiency and material selection. Several interviewees also point out that the main driving forces are compliance and competitiveness, not goodwill. Interestingly, the case companies are very different as to how sustainability initiatives are driven: while company C is very much top-down steered, at companies A and B it is more of a bottom-up commitment by interested product developers. The interviewees stress, however, that this engagement at the operational level is not allowed to cause any additional costs. All companies have a strategic plan

that includes some environmental and/or social aspects, but their success in communicating these documents throughout the organization is varying.

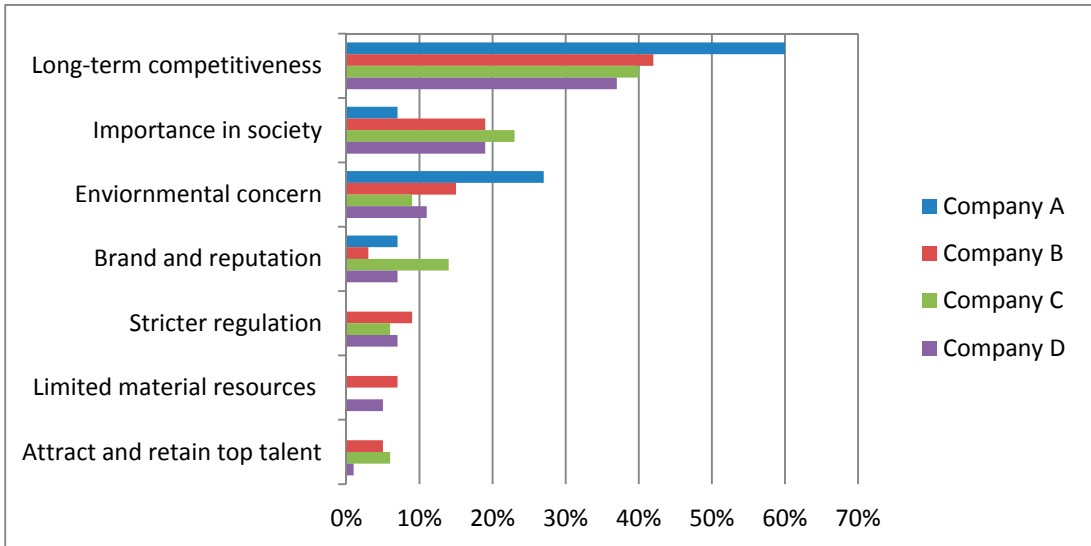

**Figure 3.** Questionnaire results for why the respondents think that increasing capabilities for sustainability implementation are important for the future of the company.

### 4.1.2. Company Challenges and Improvement Suggestions for Sustainability Integration

Several existing studies have investigated and reviewed the challenges that companies face when trying to integrate a sustainability perspective in product innovation [40–42]. The results of this study confirm that costs and short-term economic thinking are perceived as a main barrier: "*The challenge is often to keep down the costs of the final product*". Frequently stated reasons are that: (i) the relation between increased sustainability performance and profitability is unclear; and (ii) the cost of the product might get higher and it is unclear if customers are willing to pay extra for a more sustainable product. In the short term, there might also be (iii) higher investment costs; and (iv) a risk of lower profitability. One questionnaire respondent at Company D states that "*More sustainable solutions are welcome, but they are not allowed to cost more.*" Considering that the main driving force for sustainability implementation was described in terms of increased profitability and competitiveness, these results appear to be contradictory. This paradox was also part of the results of Høgevold et al. [9], in which interviewees point out the economic rationale behind sustainability proactivity, while at the same time highlighting the challenge of being sustainable and profitable at the same time. The results of our study raise the possibility of two possible explanations. First, sustainability is not yet thought of as an integrated part. Instead, many respondents still consider it to be a separate goal that requires time and resources in order to be achieved. This leads to a perceived conflict between sustainability goals and financial goals. However, sustainability should rather be treated not only as a goal in itself, but also as a means to achieve other goals, including financial ones [30]. This means that sustainability and profitability may exist in symbiosis: sustainability drives profitability, which in turn can lead to more investment in sustainability initiatives. Second, the time perspective can be a critical point: a strong sustainability profile might cause some costs and risks in the short run, even if it is a necessity for survival and profitability in the long run. Also, most of today's incentives are focused on short-term economic performance. Other frequently stated challenges are (i) lack of sustainability criteria to decide on what the most sustainable solution is; (ii) "*Commitment from the top management feels non-existent [ . . . ]*"; (iii) control over sustainability aspects throughout the supply chain; (iv) limited possibilities to influence; and (v) lack of knowledge; (vi) "*Complex products with long lifetime can be hard to analyze from a sustainability perspective.*"

The most stated improvement suggestion was better or other support tools, i.e., support tools that can provide a comprehensive overview of the sustainability impact of design alternatives and also provide guidance in trade-off situations. These support tools should include a full sustainability perspective, as the focus so far is only on material selection and energy efficiency, which is in line with existing work [34,42]. Higher management commitment, together with education and training, were also frequently mentioned improvement suggestions. The descriptive studies revealed that "*the idea of sustainability is indeed spread, but it lacks the details [ . . . ]*". However, this is an important pre-requirement for effective communication and all sustainability-related work. At company D, about half of the respondents had participated in a sustainability training, while the other half had not. Results show that employees who had received training scored significantly higher ($p = 0.02$) on how important they think that sustainability integration for the future success of the company is. They also both know of and use more formal decision support tools for sustainability ($p < 0.01$). Even though these findings are based on only one company, they still provide an interesting empirical indication of the positive effect of sustainability training, as previously suggested by Bansal and Roth [43].

### 4.1.3. Sustainability Capabilities and Maturity Degree on the Organizational and Strategic Level

In total, all companies have a rather high maturity level at the strategic level according to the self-assessment results. Results from the TADS show that the companies have integrated sustainability into most of their general strategic work. However, this commitment is not always communicated successfully within the whole organization, which is a weakness according to Székely and Knirsch [44] and Boks [45], who state that communicating sustainability, e.g., investments and achievements, is important to show the seriousness of the company intent for both internal and external stakeholders.

The EcoM2 shows that the degree in which the companies have formalized their approaches to work with sustainability within different areas varies. Most companies have rather formalized approaches for the integration of sustainability on the strategic level but less formalized approaches regarding the more concrete, product-related sustainability issues as well as in the consideration of environmental trends. According to Pigosso et al. [32], the companies need to improve their documentation of the processes and the infrastructure of responsibilities and resources to support the sustainability integration in the product innovation process in order to increase their maturity level for sustainability implementation. Based on the results from Willard's [7] sustainability compliance level, the companies' maturity levels vary mostly between level 2, 'elementary', and level 4, 'outstanding'. Their average maturity degree lies around level 3, 'satisfying'. According to the method of Baumgartner and Ebner [31], all the case companies have high or very high maturity levels in the social categories while at least some of the companies have significantly lower maturity for the ecological and economic aspects. From plotting the level for sustainability aspects, different maturity levels were visualized. This shows that company A has the highest and most even results of the three companies, mostly between levels 3 and 4. For company B and company C, the results are varying more between the categories and aspects, see Figure 4.

The combined results of the methods revealed four main challenges for the case companies. Firstly, all companies face a challenge in implementing existing sustainability strategies in practice on operational level. This challenge is strengthened by the results from the questionnaire and interview study. Secondly, there is a weak and unclear internal communication of visions, goals, and strategies throughout the organizations, especially the communication between the senior management and the other employees. Thirdly, the formalized consideration of environmental trends is weak in these organizations, and fourthly, the economic and ecological sustainability aspects need improvement in order to reach the same maturity levels as the internal and external social sustainability aspects.

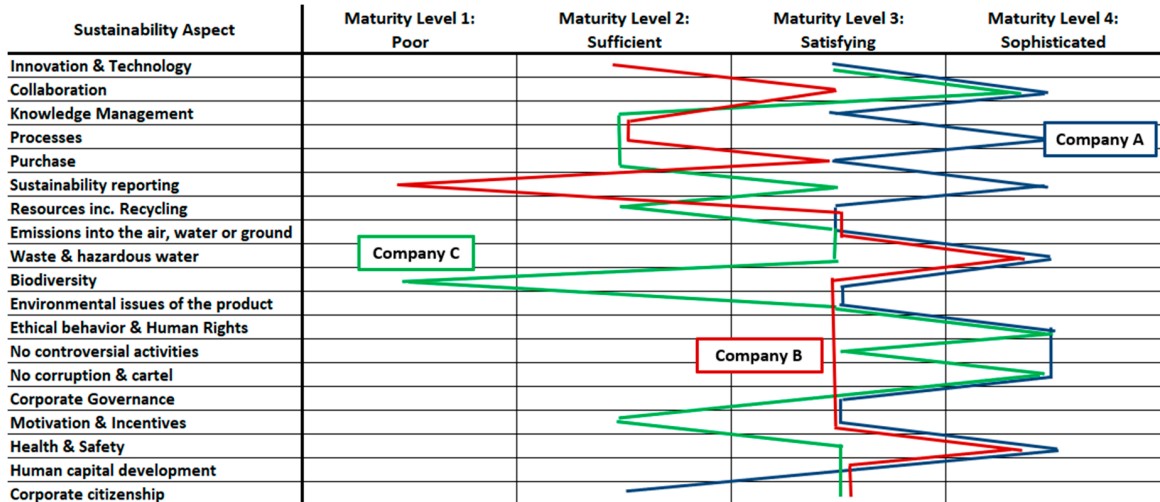

**Figure 4.** The results from the self-assessments plotted as lines along the different sustainability aspects to get strategy profiles of the companies as described by Baumgartner and Ebner [31].

## 4.2. Experiences from the Practical Application of Existing Methods for Maturity Assessment

The results of the different methods are coherent and complement each other. The TADS templates are valuable as the guided questions in the templates capture qualitative information that cannot be gathered with the other methods. Furthermore, the method gives a good indication of how well sustainability is integrated into the general business organization on the strategic level. An identified problem of the TADS is that the managers, who did the self-assessment, in many cases have difficulties to elaborate on 'how' something is done. A possible solution could be to ask supplementary questions in order to get more information.

Willard's sustainability compliance ladder can be seen as a rough indication of how far the companies have come on the way towards full sustainability integration. However, the method allows for subjective interpretation and is therefore not suited to give a certain or detailed result. The EcoM2 adds information on how formalized the companies' approaches concerning sustainability are, which is crucial for assessing their maturity levels. The method also contains questions on the strategic work as well as questions that are more product-related. It revealed that the companies generally have a lower maturity level when it comes to considering environmental trends.

The templates that were based on the work by Baumgartner and Ebner [31] added valuable results as they had a higher level of detail and as they revealed how different sustainability aspects and categories have different levels of maturity in the companies. The method could be further improved by substituting the maturity levels and checkboxes with continua which would enable the persons who do the self-assessment to indicate an opinion of the company being in between two levels. The attempt to see if and which strategy the companies follow did not result in any good match. According to Baumgartner and Ebner [31], this might indicate that the companies pursue sustainability more coincidentally rather than with a clear strategy.

From the experiences of applying the four methods for maturity assessment, some interesting findings and improvement suggestions were concluded. Firstly, the methods used in this study give coherent results and complement each other. At the same time, there is some overlap and discrepancies between the methods, e.g., different definitions of maturity levels. Secondly, in self-assessments, the subjectivity needs to be handled in a way that minimizes the bias in the results and at the same time does not require too many resources. Thirdly, future development should consider how qualitative information on 'how' the companies work with different sustainability aspects can be gathered without requiring confidential information.

*4.3. Research Stage 2: Development of a Self-Assessment Method of Capabilities for Sustainability Implementation in the Product Innovation Process (SAM4SIP)*

Based on the application of different existing sustainability assessment methods, the following criteria were identified as most important for creating industrial value: (i) be applicable by companies without external help; (ii) generate results that have a good balance between being general and concrete; (iii) be applied with reasonable time and resources; (iv) measure the level of current capabilities for sustainability implementation; (v) handle the subjectivity of the self-assessments in a way that minimizes the bias in the results and at the same time does not require too many resources; (vi) be applicable to build a roadmap for systematic improvement; and, (vii) be supportive as a communication tool at the same time as being able to measure the progress. None of the existing methods fulfills all criteria, see Table 3. The combination of methods, like in research stage 1 of this study, requires too much time and resources and the results are difficult to consolidate and use. Guided by the criteria, a new version of a self-assessment approach, SAM4SIP, was developed to provide a systematic and efficient approach to support sustainability implementation.

**Table 3.** A summary of the strengths and weaknesses of the four assessment methods based on insights from the practical application.

| | **The Five Levels of Sustainability Integration [30]** | **Company Sustainability Strategies [31]** | **Eco-design Maturity Model [32]** | **Template for Assessing Decision Systems [13]** |
|---|---|---|---|---|
| **Aim** | To describe five levels of sustainability integration in corporations. | To assess the consistency in the implementation of a certain sustainability strategy. | To improve the processes related to product development within th company. | To identify key improvements for how to integrate sustainability in the strategic decision system. |
| **Strengths** | Based on case studies of front-running companies. Gives a quick indication of how far the companies have come on the way towards full sustainability integration. | Covers ecological, social and economic dimensions of corporate sustainability. Includes detailed sustainability aspects at different maturity levels. | Questions on the strategic work as well as questions that are more product-related. Information on how formalized approaches concerning sustainability are. | Includes a strategic long-term sustainability perspective. Captures descriptive qualitative information that gives a good indication of how well sustainability is integrated on the strategic level. |
| **Weaknesses** | Does not include details regarding aspects for the different levels. Rough assessment not suited to give detailed results. | Results of the templates and the strategies are difficult to interpret. | Requires expert knowledge and support from a consultancy. Does not cover the social dimension. | Hard to capture 'how' something is done as detailed questions are lacking. |

4.3.1. Defined Templates with Sustainability Compliance Index Levels for Each Key Element Aspect

The previously identified key elements for successful implementation of a sustainability perspective in the early product innovation process [19] clarify the areas that are relevant to consider and were, therefore, used as a base in the development of SAM4SIP. These key elements are, however, rather general and do not provide guidance for companies on what aspects to consider in practice and how to improve the implementation of sustainability in the product innovation process systematically. Through the application of the four different self-assessment methods, detailed aspects for each key element were discovered, see Table 4. The maturity characteristics for the aspects were developed through the use of an adapted version of the sustainability compliance index (SCI) levels defined by Hallstedt [46].

**Table 4.** Newly-derived, detailed aspects that are relevant for each key element.

| Existing Key Elements | Newly-Derived, Detailed Aspects of Each Key Element to Assess against—and Examples of What Those Mean When a Company Has Reached the Highest Maturity Level | | | |
|---|---|---|---|---|
| 1: Ensure organizational support from senior management to integrate sustainability | Senior management commitment, e.g., sustainability permeates the whole organization at all levels from strategic level, tactical level to operational level. | Strategic sustainability plan, e.g., formulated goals and strategies for long-term future product development. | Communication— e.g., effective external communication of commitment and sharing of experiences. | |
| 2: Sustainability perspective early in the product innovation process | Sustainability in all life-cycle phases— e.g., all relevant ecological as well as social aspects in all product life-cycle stages are considered in a systematic way. | When in the product innovation process sustainability aspects are considered— e.g., sustainability related decisions are predominantly taken during the early phases of product development. | How sustainability aspects are considered in the early phases— e.g., sustainability aspects are fully integrated into both processes and decision support tools. | |
| 3: Utilize knowledge and experience of procurement staff in the earliest phases of the process | Involvement of procurers— e.g., procurers are systematically involved in the earliest phases of product development and throughout the product innovation process. | Supplier requirements—e.g., all relevant ecological and social aspects are included in supplier evaluations as strict requirements for suppliers. | Supplier assessments and audits— e.g., assessments and audits are regularly and systematically performed at suppliers and actions are taken. | Supplier relationships— e.g., the company is building long-term relationships with its main suppliers in order to continuously improve both the company's own and the suppliers' sustainability performance. |
| 4: Consideration of social aspects across the product life-cycle and its value chain | Which social aspects are considered—e.g., health; influence; competence; impartiality; and, meaning- making are systematically considered in all product life-cycle stages and across the company's value-chain. | In which ways social aspects are considered—e.g., social aspects are both a central part of policy and steering documents, as well as of processes and support tools. | | |
| 5: Assigned responsibilities for sustainability implementation in the product innovation process | Roles and responsibilities—e.g., there is a designated role that is responsible for sustainability implementation in the product innovation process. | | | |
| 6: A systematic way of competence building | Education—e.g., all employees have gone through at least a basic education and training in sustainability. | Knowledge sharing & management—e.g., knowledge is well documented, stored, and conveniently and extensively shared and applied. | Follow-up and lessons learned—e.g., thorough and systematic follow-up of projects is performed and the lessons learned are actively and systematically used to improve future decision-making. | Sustainability understanding—e.g., there is a clear and well-communicated definition and understanding of what sustainability means, which is shared by all employees. |
| 7: Utilize tools for guiding decisions in product development | Use of formal decision support—e.g., formal decision support tools for sustainability aspects are systematically and actively used to improve decision making in early phases of the product innovation process. | Use of informal decision support—e.g., a culture of effective use of informal decision support tools, such as discussions with colleagues. | Sustainability perspective in decision support—e.g., decision support tools include a full socio-ecological sustainability perspective. | |
| 8: Utilize tools that incorporate a backcasting perspective from a definition of success | The approach— e.g., a backcasting perspective is well-integrated into processes and support tools on strategic, tactical, and operational levels. | The vision— e.g., the vision includes a full socio-ecological sustainability perspective and provides a clear and strong basis for backcasting. | | |

In detail, a top-down approach was applied, using backcasting from an ideal vision on all aspects, to give answers to how companies ideally would need to work in relation to the identified aspects to be compliant with the sustainability principles [20]. The empirical data from the descriptive studies was then used to calibrate the SCI levels for each aspect. Figure 5 shows the different development steps of SAM4SIP.

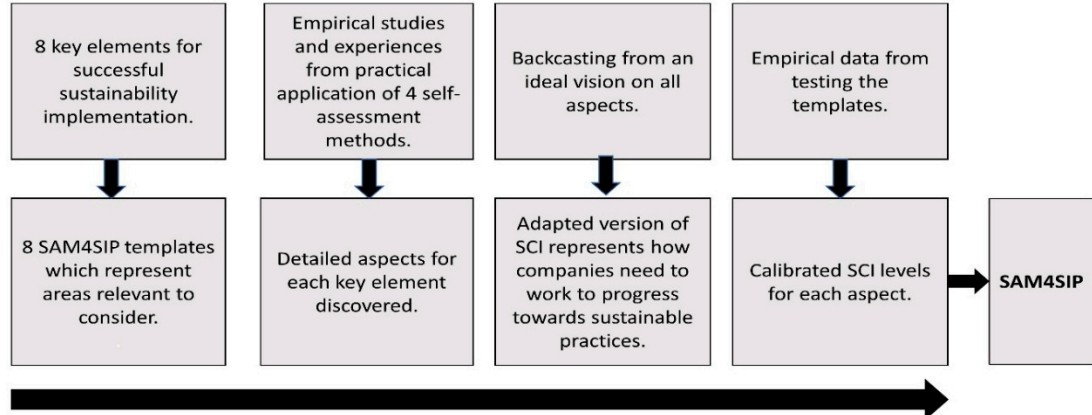

**Figure 5.** The development steps of the Self-Assessment Method of capabilities for Sustainability Implementation in the Product innovation process (SAM4SIP).

### 4.3.2. The Proposed Self-Assessment Method—SAM4SIP

The purpose of the proposed self-assessment method, SAM4SIP, is to gain insight on how the company currently performs in relation to the eight key elements for successful sustainability integration in the product innovation process. This means that areas can be identified that are already well-developed or need further improvement. Thereby, a baseline is established, which can be used to focus resources and improvement efforts on the areas that need them the most and that yield the greatest return. Also, the template can provide guidance for how to move forward, build a roadmap and measure the progression of the sustainability integration and implementation in the product innovation process. This tool can also provide a benchmarking between companies, or between different units within a larger global organization, and thereby measure the different maturity levels for the organization.

Practically, the tool includes guidance for how to use the method, definitions of some terms used in the templates, and an example of how to fill in the templates. The first step is to read and discuss each aspect in the company team and then indicate what level the current performance of the company best corresponds to. As companies are complex organizations, it is difficult to have a complete picture and to make a precise assessment for all aspects. There is, also, a risk of subjectivity in doing a qualitative assessment. Therefore, the users should indicate the level of certainty for each assessment on a scale of 1–10, with 1 meaning a very uncertain assessment judgment and 10 a very certain judgment. For each aspect, there is also a possibility to explain in short what the assessment is based on or to refer to, for example, a document or one's own experience. One of the templates, representing key element 1, is presented in Figure 6. The whole set of templates can be found in the Supplementary Material.

| Assessed by: | | Role: | Date: | |
|---|---|---|---|---|

| **Key Element 1: Ensure organizational support from senior management to integrate sustainability** | | | | |
|---|---|---|---|---|
| **Sustainability Compliance Index (SCI) Level** | | **Senior management commitment** | **Strategic sustainability plan** | **Communication** |
| 9 | Sophisticated/ outstanding | There is clear and visible commitment from senior management to include a socio-ecological sustainability perspective in the company's mission and vision and in its core values. Sustainability permeates the whole organization at all levels from strategic level, tactical level to operational level. This requires commitment all the way from the top management via managers to the individual employee. | The strategic plan including sustainability integration is the main driver for sustainability improvements. The plan incorporates and connects all relevant aspects and areas. Clear routines are in place for regular updating and follow-up. There are formulated goals and strategies for long-term future product development that include the complete product life-cycle. | Both the commitment and the strategic sustainability plan are well-communicated at all levels of the company and known by all employees. Clear routines are in place for information spreading and dialogues for learning and sharing regarding sustainability issues. Effective external communication of commitment and sharing of experiences. |
| 8 | | | | |
| 7 | | | | |
| 6 | Satisfying | There is a commitment from senior management to include a socio-ecological sustainability perspective in the company's strategic, tactical and operational levels. However, there is still some improvement potential: to formulate a complete socio-ecological persepctive in one document, to strengthen the top-down approach, and to make sustainability permeate the whole organization as a core value. | A strategic plan including sustainability integration exists but does not cover a complete socio-ecoligical sustainability perspective. The connection to tactical and operational level is still vague. | The strategic sustainability plan is well-communicated at all levels of the company and known by all employees. Dialogue- and information sharing teams with focus on sustainability issues are encouraged by management. Commitment is communicated externally on non-regular basis. |
| 5 | | | | |
| 4 | | | | |
| 3 | Elementary | A few systematic sustainability activities are supported from management. Still stronger bottom-up initiatives with some support from management. Partly commitment only from individual persons in management. There are both social- and ecological aspects that are considered but they are not formulated and connected to each other and the to the organization's purpose and operations. | There is no distinct strategic plan including sustainability integration but some document that could be interpreted as sustainability plan exists and is used at the company. It does however not adress and connect the strategic, tactical, and operational levels of the organization. | There is a systematic way to spread information and communicate internally regarding sustainability issues. Dialogues and information-sharing with focus on sustainability issues are done in an non-regular way. Commitment is rarely communicated externally. |
| 2 | | | | |
| 1 | Beginning | Uncoordinated, non-systematic sustainability activities. There are some bottom-up initiatives that try to influence management. Partly commitment only from individual persons in management. Only fragments of sustainability are considered. | There is neither a distinct strategic plan including sustainability integration nor any other documents that include a strategic sustainability perspective. | Lack of processes and routines for communication around sustainability goals and plan. There are no formal ways of information-spreading and no dialogues or teams for sharing information and knowledge. Commitment is not externally communicated. |
| How certain are you regarding your assessment/answer? | | Very uncertain — Very certain 1 2 3 4 5 6 7 8 9 10 | Very uncertain — Very certain 1 2 3 4 5 6 7 8 9 10 | Very uncertain — Very certain 1 2 3 4 5 6 7 8 9 10 |
| Number: | | | | |
| Assessment based on / Reference to document or other material | | | | |
| Comment | | | | |

**Figure 6.** An example of one of the SAM4SIP templates, key element 1.

## 4.4. Research Stage 3: Validation and Improvement of SAM4SIP Method

The research team applied the SAM4SIP to assess the sustainability capabilities of companies A, B and C. This test resulted in several improvements of the SCI-level descriptions and the calibration. Overall, the companies were found to be mainly between the "beginning" and "satisfying" SCI levels, which is in line with the results of the other applied self-assessment methods. However, the methods assess different things and the unique value of SAM4SIP is that it provides a more detailed picture in relation to the eight key elements and sub-aspects. Thereby, it ensures that all relevant areas, including both environmental and social dimensions, are covered. It is also designed to be easy to use and communicate and to provide guidance for systematic progression towards increased sustainability capabilities, especially on the strategic and tactical levels. The real-life application of the tool at companies B and C showed that the companies assess themselves very similarly to the researchers' assessments. The results of the companies' self-assessment are summarized in Figure 7. For example, the team at Company B rated aspect 7, "involvement of procurers" to be on SCI level 4, because they

think that processes are in place to involve procurers early on in the product innovation process, but they consider the connection to sustainability to be insufficient. However, they need to gather more information, as they felt unsure (certainty level 4) about the assessment. The lack of a shared understanding of sustainability (aspect 17) and the absence of a clear vision to do backcasting from (aspect 22), were identified as important improvement areas.

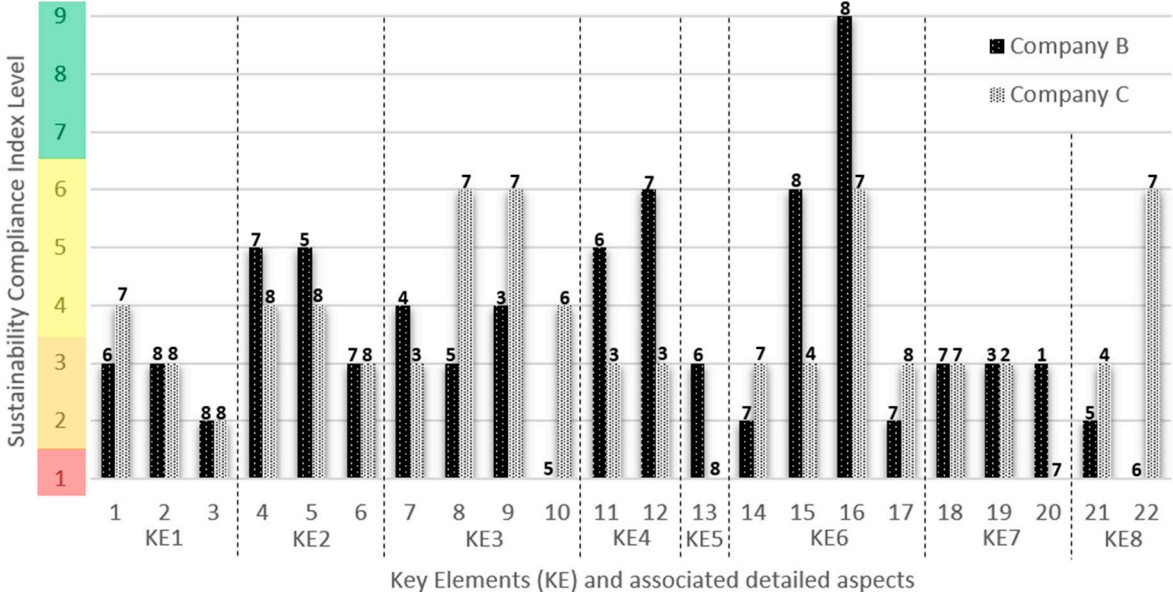

**Figure 7.** Overview of the companies' SAM4SIP results. The numbers above the bars indicate the companies' perceived level of certainty regarding the assessment.

The follow-up workshop session, including the validation questionnaire, provided valuable feedback. The self-assessment tool was considered to be most useful for (i) getting an overview of the current state of sustainability capabilities; (ii) identifying areas of strengths and weaknesses; (iii) providing guidance and creating a roadmap towards increased sustainability capabilities; (iv) identifying improvement actions; (v) starting conversations about sustainability within the company; and (vi) creating management attention. It was found to be useful for benchmarking to some degree but not very useful for clarifying the direct business case of improved sustainability capabilities. However, the main objective of the tools is to guide systematic improvement and not to provide quantifiable evidence for the business benefits. The self-assessment approach of the tool was appreciated, and the companies were able to use it without external help. Drawbacks were also pointed out, for example, the subjective character of the assessment and the fact that some terms might be interpreted in different ways. Barriers to implementation of the tool were stated as the limited access to the people that need to fill it in and the fact that the assessment takes about two hours in total. Improvement suggestions included: (i) a clarification of the scope of the assessment and some key terms; (ii) developing a simplified version that takes less time and that can be sent out to many people in a questionnaire format to get the views of many different roles and functions at the company, complementing the detailed assessment; (iii) improved visualization of results to ease communication.

All participants were positive to further testing and developing the tool. Since then, Company B has independently continued to use the tool and implemented it as part of their roadmap development work. A group in Company C is using the tool mostly to create management commitment and to anchor sustainability initiatives within the company.

## 5. Conclusions

The systematic implementation of a sustainability perspective in product innovation and the continuous improvement of related practices, requires an efficient approach that enables companies to assess their current maturity level and that provides guidance for progression over time. In this study, a questionnaire and interviews were conducted, and several existing sustainability maturity assessment methods were applied at Swedish product development and manufacturing companies. The findings showed that none of the existing methods alone can fulfill the needs of industry practitioners in terms of required time, level of detail, etc.

To fill this gap, a new method, SAM4SIP, was developed and tested in real cases. It provides companies with a clear overview of their current capabilities for sustainability implementation in the product innovation process in relation to eight key elements for success. Thereby, areas of strengths and weaknesses can be identified, and efforts can be focused on the leverage points with the greatest improvement potential. Comparing the current capabilities with the characteristics of higher maturity levels provides insights on what actions and measures the company must take to systematically improve its capabilities for sustainability implementation. Ideally, this work is integrated with existing roadmap processes. Positive feedback indicates the usefulness of the presented method. In contrast to the other self-assessment methods, the suggested SAM4SIP is specifically focused on sustainability implementation in the product innovation process and the outcome gives concrete measures and actions to support the development of more sustainable solutions from the company. Another added value is the long-term perspective, which means that the ideal capability level to implement sustainability for each aspect is described.

It was validated that SAM4SIP is applicable by the case companies without external help and that it can be applied with reasonable time and resources. Independent and continuous use by the companies of the developed self-assessment method beyond this particular study provide an indication of the practical value of the results. Even though SAM4SIP was developed to be applicable by a wide range of product development and manufacturing companies, further investigations and tests are necessary to validate the applicability and usefulness of the method in different contexts, such as different industries, countries, and company sizes. Another limitation of this study is that not all existing self-assessment methods for sustainability maturity were tested.

Besides the prescriptive contribution to practice, this study also contributes to knowledge. Firstly, by further specifying detailed aspects that need to be considered for each of the eight key elements for successful sustainability implementation in product innovation. Secondly, the study provides insights on the strengths and limitations of several existing sustainability maturity models through application in real cases.

Future work will include (i) further validating and testing at more companies, or different units or sites within a larger organization, also allowing for benchmarking; (ii) research on how the template can be linked to key performance indicators [47], providing a more tangible connection to the business benefits of increased capabilities for sustainability integration and implementation; and (iii) investigations into how the results can be visualized in a way that gives a quick and easy overview of the current state to understand, communicate and to create improvement plans. The detailed SAM4SIP templates will also be complemented with a short version of the tool, so that a larger number of employees at different units and departments can do the assessment, which would provide first-hand input for the focus group that undertakes the more detailed assessment.

**Supplementary Materials:** The following are available online at http://www.mdpi.com/2071-1050/10/12/4336/s1, complete SAM4SIP templates.

**Author Contributions:** Conceptualization, J.S. and S.I.H.; Data curation, J.S.; Formal analysis, J.S.; Funding acquisition, S.I.H.; Investigation, J.S. and S.I.H.; Methodology, J.S. and S.I.H.; Project administration, S.I.H.; Supervision, S.I.H.; Validation, J.S. and S.I.H.; Visualization, J.S. and S.I.H.; Writing—original draft, J.S. and S.I.H.

**Funding:** This research and the APC were funded by Sweden's Innovation Agency, grant number 2018-00361, and the Knowledge Foundation in Sweden, grant number 20120278.

**Acknowledgments:** Financial support from the Knowledge Foundation and Vinnova in Sweden is gratefully acknowledged. Sincere thanks to the industrial and academic research partners for valuable advice.

**Conflicts of Interest:** The authors declare no conflict of interest. The founding sponsors had no role in the design of the study; in the collection, analyses, or interpretation of data; in the writing of the manuscript; and in the decision to publish the results.

## Appendix A

Questionnaire questions:

1.  What is your role/working tasks? (single choice) Development Engineer, Project Leader, Specialist, Manager, Staff, Purchasing, Other:
2.  How many years have you been working at the company? (single choice) 1–5, 6–10, 11–15, 16–20, 21–25, 26–30, More than 30
3.  Where in the innovation process do you mainly work? (multiple choice) Feasibility study, Experimental testing and analysis, Technology development, Product development, Industrialization, Maintenance/Service, Other:
4.  Have you participated in the Health, Safety and Environment (HSE) Sustainability Awareness Training? (only Company D) Yes, No
5.  To what degree does the company prioritize sustainability in its day-to-day activities? Scale: 1 (very low)–10 (very high)
6.  To what degree does the company prioritize sustainability at a strategic level? Scale: 1 (very low)–10 (very high)
7.  To what degree do you think an increased capability to integrate sustainability aspects is important for the future of the company? Scale: 1 (not important at all)–10 (very important)
8.  Why? Motivate your answer in previous question.
9.  What formal support for decision-making do you know regarding sustainability issues at the company?
10. Which of these formal decision supports do you use?
11. What informal decision support do you use?
12. How good are the decision supports at the company? Scale: 1 (very bad)–10 (very good)
13. Do you think that the available support addresses both environmental and social aspects of sustainability? If yes: give examples. If no: what do you think is missing?
14. In your opinion, within which areas at the company are decisions that affect sustainability taken? (multiple choice) Research projects, Consultancy projects, Experimental planning and execution, Technical support, Purchasing, HSE, Management/Strategy, Other:
15. When in the innovation process at the company do you think decisions that impact sustainability are taken? (multiple choice) Project proposal stage, Project requirement stage, Feasibility study, Technology development, Product development, Experimental testing and analysis, Industrialization, Maintenance/Service, Other:
16. Who at the company makes the decisions regarding sustainability? (multiple choice) Development Engineers, Project leaders, Specialists, Managers, Local Sustainability Officer Purchasers, Human Resources (HR), Other staff, Do not know
17. Specify which sustainability aspects (in the process of product innovation) the company takes decisions about (in your opinion)?
18. What is the driver and/or the requirements behind the decision regarding sustainability at the company?
19. What challenges/problems do you face at the company when making a decision regarding sustainability issues related to company products or services?
20. What improvements would you like to see at the company in terms of support for sustainability issues?

21. Is there something else you would like to add regarding sustainability, specifically in relation to product innovation?
22. To what age group do you belong? (single choice) Younger than 30, 31–40, 41–50, 51–60, older than 60

**Appendix B**

Interview questions in relation to the eight key elements.

**Key Element 1**

1.1 Is there a commitment from senior managers to integrate sustainability (a) in the company? (b) in product development?
1.2 Do you have a strategic sustainability plan that is well communicated at the company?

(a)     If yes: how was the sustainability plan developed and who was involved in that?
(b)     Could you share this plan with us?

**Key Element 2**

2.1 Do you bring in a sustainability perspective in the product innovation process?
2.2 If yes, when do you bring in a sustainability perspective in the product innovation process?
2.3 If yes, how do you align the sustainability perspective throughout the design process?
2.4 How far in the sustainability integration do you think the current stage is? (i) in the beginning (ii) half way (iii) almost complete.

**Key Element 3**

3.1 Does the company develop sustainability requirements for suppliers?
3.2 When do the procurers work in the product development process?
3.3 Can you specify if, and in that case how, the procurers could influence decisions from a sustainability perspective?

**Key Element 4**

4.1 Do you consider social aspects across the product life and its value chain?
4.2 If yes, what aspects do you consider?
4.3 If yes, how do you consider these?
4.4 If no, why not?

**Key Element 5**

5.1 Can you list the roles for people responsible for sustainability implementation in the product innovation process at your company?
5.2 If you have no specific role for that, can you explain how this area is taken care of?

**Key Element 6**

6.1 Do you have a systematic way to build competence in this area?
6.2 If yes, does the competence building include (a) knowledge sharing? (b) follow up actions; and (c) re-use of evaluations to increase the competence in the sustainability field?

**Key Element 7**

7.1 Do you have support tools for guiding decisions (e.g., checklists, guided questions, criteria) as a complement to assessment tools (e.g., life-cycle assessment) regarding sustainability aspects in product innovation and development?

**Key Element 8**

8.1 Do you use tools that incorporate a backcasting perspective from a definition of success in the product innovation process?
8.2 Do you include sustainability in this definition of success?

**Appendix C**

**Table A1.** Template based on the Eco-design Maturity Model (EcoM2) that was used as a self-assessment in order to get an indication of the employees' perception of the company's capability levels regarding different aspects.

| | Questions | Level 1–5 |
|---|---|---|
| 1. | How formalized is your approach to formulate and update the company's environmental policy and/or strategy? | |
| 2. | How formalized is your approach to deploy and maintain an environmental policy and/or strategy in the product level? | |
| 3. | How formalized is your approach to effectively integrate product-related environmental goals into the corporate strategy? | |
| 4. | How formalized is your approach to integrate the environmental dimension in the strategical decision-making process jointly with the traditional aspects? | |
| 5. | How formalized is your approach to establish product-related vision, strategy and environmental roadmaps in the strategic level at the company? | |
| 6. | How formalized is your approach to strategically consider the product environmental performance in the company portfolio management? | |
| 7. | How formalized is your approach to develop business, product and market strategies considering the environmental trends? | |
| 8. | How formalized is your approach to incorporate product-related environmental goals into the technological strategy? | |
| 9. | How formalized is your approach to define a strategic roadmap for the development and implementation of new technologies that allow a better environmental performance over the product life cycle? | |

**Appendix D**

Questions used in the validation session of SAM4SIP with company practitioners.

**Applicability and usability**

1.  Do you think the tool can give support in measuring progression towards sustainability implementation in the organization for sustainable product development?

    a.  If yes—how?
    b.  If no—why not?

2.  To what extent (scale 1–10) could the self-assessment tool be beneficial for:

    a.  Gaining an overview of the current state of sustainability capabilities?
    b.  Identifying areas of strengths and weaknesses?
    c.  Clarifying the business benefits of increased sustainability implementation?

      d.     Identifying actions to improve sustainability capabilities?

      e.     Providing guidance and creating a roadmap towards increased sustainability capabilities?

      f.     Potential benchmarking with other companies or other units within the company?

3.    Beneficial for other things? What in that case?

4.    Do you see any disadvantages or risks with using it?

5.    Do you see any improvement potentials of the suggested approach?

**User—role**

6.    Do you know if you conduct similar sustainability assessments at the company today?

      a.     If yes—what tools are used in that case?

      b.     If yes—who is responsible for this now?

      c.     If not, why not?

      d.     If not today, what role(s) at the company should do this and could be a potential user of the self-assessment tool?

**Self-assessment approach**

7.    What advantages with using a self-assessment approach do you see?

8.    What disadvantages with using a self-assessment approach do you see?

9.    Do you have any ideas for how to overcome these disadvantages?

**Further testing, implementation, and validation**

10.    Would it be interesting to further develop, apply, and demonstrate the tool and its impact in other parts or units of the company?

11.    Do you see any barriers for implementation of the tool and, in that case, which?

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
