# Peer review of "Self-Assessment Method for Sustainability Implementation in Product Innovation"

_sustainability, doi:10.3390/su10124336_

Reviewer 1 Report

Abstract: it would be wise to also highlight the theoretical contribution of your paper, i.e. how does your paper enhance the literature. Which research gap is your paper covering?

Introduction: The introduction should be not merged with the literature review. You should very clearly make a delimitation between the literature review and the introduction. You would need to focus in the introduction on the research framework, the theory you are grounding your research on, the novelty of the research and explain how the research question transposes in the methodology you use and the results that you obtain. Furthermore it should be very clear explained is this is really an original research or if it is only an extension of previous research, as you also point out that you are basing your research on previous descriptive work.

You should end the introduction with a paragraph where you describe the sections of your article.

Literature review: is mixed with the introduction. It is not very common for research articles. The two sections should be treated separately.

Methodology: The research design of the article is interesting, however there is no operationalization of the variables.How did you come up with the items / questions from the appendix? 

How was the reliability and validity and internal consistency of the data checked? From where did you know that your data is trustworthy? You have not performed any statistical analysis to validate the answers you got. I doubt that the findings based on the questionnaire are really representative, as you did not employ any sampling plans.

Discussions: The results are interesting, however it would be necessary to validate them also for a specific industry. 

Conclusions: It is not clear what the real added value to the knowledge is. Furthermore some limitations should be also considered.

Author Response

Please find our response in the attached file.

Reviewer 2 Report

 Lits of literature lasks the latest literature in the field.

Such questions as information availability for sustainability management, social responsibility, emerging new technologies, technology transfer through clusterin are not considered. Therefore research limitations have to be formulated

Please find sources, which would enrich your paper. It is recommended to incorporate them.

Vegera, S.; Malei, A.; Sapeha, I.; Sushko, V. 2018. Information support of the circular economy: the objects of accounting at recycling technological cycle stages of industrial waste, Entrepreneurship and Sustainability Issues6(1): 190-210. https://doi.org/10.9770/jesi.2018.6.1(13)

Mishenin, Y.; Koblianska, I.; Medvid, V.; Maistrenko, Y. 2018. Sustainable regional development policy formation: role of industrial ecology and logistics, Entrepreneurship and Sustainability Issues 6(1): 329-341. https://doi.org/10.9770/jesi.2018.6.1(20)

Kantemirova, M. A.; Dzakoev, Z. L.; Alikova, Z. R.; Chedgemov, S. R.; Soskieva, Z. V. 2018. Percolation approach to simulation of a sustainable network economy structure, Entrepreneurship and Sustainability Issues 5(3): 502-513. https://doi.org/10.9770/jesi.2018.5.3(7)

Benešová, D.; Kubičková, V.; Michálková, A.; Krošláková, M. 2018. Innovation activities of gazelles in business services as a factor of sustainable growth in the Slovak Republic, Entrepreneurship and Sustainability Issues 5(3): 452-466. https://doi.org/10.9770/jesi.2018.5.3(3)

Oganisjana, K.; Svirina, A.; Surikova, S.; Grīnberga-Zālīte, G.; Kozlovskis, K. 2017. Engaging universities in social innovation research for understanding sustainability issues, Entrepreneurship and Sustainability Issues 5(1): 9-22. https://doi.org/10.9770/jesi.2017.5.1(1)

Monni, S.; Palumbo, F.; Tvaronavičienė, M. 2017. Cluster performance: an attempt to evaluate the Lithuanian case, Entrepreneurship and Sustainability Issues 5(1): 43-57. https://doi.org/10.9770/jesi.2017.5.1(4)

Tetsman, I.; Bazienė, K.; Viselga, G. 2017. Technologies for sustainable circular business: using crushing device for used tires, Entrepreneurship and Sustainability Issues 4(4): 432-440. https://doi.org/10.9770/jesi.2017.4.4(3)

Jankalová, M.; Jankal, R. 2017. The assessment of corporate social responsibility: approaches analysis, Entrepreneurship and Sustainability Issues 4(4): 441-459. https://doi.org/10.9770/jesi.2017.4.4(4)

Author Response

(The authors gave the same response as above.)

Reviewer 3 Report

Dear Authors,

Thank you for your interesting paper. I have three recommendation that may help you to improve the paper:

I recommend you to extend your review (1.2 Maturity models for sustainability integration) to explain why we need to develop a new self-assessment method for sustainability implementation in product innovation.

2. it needs to explain more about sustainability implementation in product innovation. What do you mean? clarify it and bring also some samples based on your case companies in the results section.

3. I would like to see what is the relation between the maturity of companies practices and sustainability implementation in the product innovation in the Results section and Conclusion.

Kindest regards,

Reviewer

Author Response

(The authors gave the same response as above.)

Reviewer 4 Report

An interesting study that is both implicit and explicit. The framework introduced SAM4SIP is well developed, as there is significant research involved. The methodology is clear and efficient. It could although, in the conclusions' section, elaborate more on the European or even global perspective of the study. It is there, but not clearly revealed. Also there should be a little more emphasis on the innovation terminology and approach in the introduction. Overall a good paper.  

Author Response

(The authors gave the same response as above.)
